# Design of Secure Microcontroller-Based Systems: Application to Mobile Robots for Perimeter Monitoring

**DOI:** 10.3390/s21248451

**Published:** 2021-12-17

**Authors:** Dmitry Levshun, Andrey Chechulin, Igor Kotenko

**Affiliations:** St. Petersburg Federal Research Center of the Russian Academy of Sciences (SPC RAS), 199178 St. Petersburg, Russia; chechulin@comsec.spb.ru (A.C.); ivkote@comsec.spb.ru (I.K.)

**Keywords:** information security, physical security, security by design, design methodology, microcontroller-based system, perimeter monitoring, mobile robot

## Abstract

This paper describes an original methodology for the design of microcontroller-based physical security systems and its application for the system of mobile robots. The novelty of the proposed methodology lies in combining various design algorithms on the basis of abstract and detailed system representations. The suggested design approach, which is based on the methodology, is modular and extensible, takes into account the security of the physical layer of the system, works with the abstract system representation and is looking for a trade-off between the security of the final solution and the resources expended on it. Moreover, unlike existing solutions, the methodology has a strong focus on security. It is aimed at ensuring the protection of the system against attacks at the design stage, considers security components as an integral part of the system and checks if the system can be designed in accordance with given requirements and limitations. An experimental evaluation of the methodology was conducted with help of its software implementation that consists of Python script, PostgreSQL database, Tkinter interface and available for download on our GitHub. As a use case, the system of mobile robots for perimeter monitoring was chosen. During the experimental evaluation, the design time was measured depending on the parameters of the attacker against which system security must be ensured. Moreover, the software implementation of the methodology was analyzed in compliance with requirements and compared with analogues. The advantages and disadvantages of the methodology as well as future work directions are indicated.

## 1. Introduction

Microcontroller-based systems now are an integral part of any sphere of our activity, that is why the importance of ensuring their security is critical [1]. The consequences of failure of such systems, including associated with activities of intruders, include financial and reputational damage as well as a threat to human life and health [2]. One of the possible attack vectors is the exploitation of vulnerabilities, the presence of which in such systems is due to various factors.

Errors during the design stage of the lifecycle of microcontroller-based systems are critical, because they lead to the presence of weak places and architectural defects. Moreover, there are situations, when fixing discovered vulnerabilities is not feasible, because the companies that developed the device or its software no longer exist [3]. It means that security is not considered during the development of microcontroller-based devices, while most of the weaknesses could be prevented during the design stage.

For example, according to the SonicWall report, malware attacks with the help of microcontroller-based devices jumped 215.7% to 32.7 million in 2018 up from 10.3 million in 2017 [4]. In 2019, the attacks continued but showed a more moderate increase of 5%, according to their 2020 Cyber Threat Report [5]. According to the Palo Alto Networks 2020 Unit 42 Threat Report “98% of all device traffic is unencrypted, exposing personal and confidential data on the network” [6].

Thus, the scientific problem to be solved is the contradiction that despite the large number of vulnerabilities regularly discovered in microcontroller-based systems, there is no general methodology for such systems design. However, different techniques in this area are widely used for specific applications: robots, railway infrastructure, smart cities, etc. Those techniques can be aimed at software, hardware, individual devices and classes of systems.

The key issue of such solutions is in focusing on certain aspects of the security, ensuring their inapplicability for providing the security of microcontroller-based systems in general. For example, techniques might not take into account the strong relationship between hardware and software elements of microcontroller-based devices [7,8,9,10] or design such devices in isolation from the system they are supposed to work in [11,12].

Moreover, techniques might provide secure connection with external systems only from the designed system side [13,14,15], be bound to specific hardware, software, platforms and architectures and do not take into account limitations of microcontroller-based devices like computational complexity, energy efficiency, size and price [16,17,18,19,20,21].

Therefore, this work is aimed at developing the original methodology for the design of microcontroller-based physical security systems. Among all possible systems, only physical security systems were chosen as an area of the application, because, in such systems, during the design process, it is required to ensure not only the functionality of the system but also its security against cyber-physical attacks [22].

The contributions to the research field are as follows:an extendable set-based hierarchical relational model,an algorithm for the formation of requirements,an algorithm for the formation of component compositions,an algorithm for the design of abstract models,an algorithm for the design of detailed models, anda design methodology that combines algorithms into a single approach.

Unlike the existing solutions, the extendable set-based hierarchical relational model represents a microcontroller-based physical security system instead of representing separate microcontroller-based devices. Such functionality neutralizes the disadvantages of analogs in terms of designing devices separately from their interaction with each other. Moreover, this model is modular, extensible and hierarchical, has a strong focus on security of the resulting solution as well as considers security elements as an integral part of the designed system.

The novelty of the algorithm for the formation of requirements is in retrieving a list of system devices, communications available to them, as well as requirements for them only on the basis of system tasks, while the list of attack actions that are possible for the attacker is retrieved in accordance with the attacker’s parameters.

Unlike other solutions, the algorithm for the formation of component compositions is retrieving abstract elements and sub-elements of the designed system in accordance with the requirements, device base and already retrieved elements, while security elements are represented as abstract elements, sub-elements and recommendations for the system and its device implementation.

The novelty of tthe algorithm for the design of abstract models is in taking into account complex dependencies between systems elements, namely, their hierarchy, nesting, communications, conflicts and requirements. Moreover, this algorithm is not limited to specific platforms and architectures and because of its abstract nature reduces the number of parameters to be searched, thereby increasing the work speed of the solution.

Unlike existing solutions, the algorithm for the design of detailed models forms a step-by-step process of detailing the abstract representations of systems in accordance with the hierarchy and mutual dependencies of their elements. Moreover, this algorithm calculates the parameters of devices based on the parameters of their elements as well as the parameters of systems based on the parameters of their devices. This algorithm does not replace the abstract model of the system, but expands and complements it.

The novelty of the methodology for the design of microcontroller-based physical security systems lies in a new approach to the design, which allows combining various design techniques on the basis of hierarchical relational model transformation algorithms. Moreover, the suggested approach is modular and extensible, takes into account the security of the physical layer of the system, works with the abstract system representation and is looking for a trade-off between the security of the final solution and expended resources.

Unlike existing solutions, the methodology has a strong focus on security. It is aimed at ensuring the protection of the system against attacks at the design stage, considers security components as an integral part of the system and checks if the system can be designed in accordance with given requirements and limitations.

It is important to note that this paper represents our latest results in the investigation and development of the methodology for design of secure systems based on microcontrollers. The previous versions of this methodology were presented in [1,2]. The key part of this methodology—the extendable set-based hierarchical relational model of microcontroller-based system—was presented in [23,24]. Moreover, one of the algorithms, namely, algorithm for the design of abstract models, was presented at the 14th International Symposium on Intelligent Distributed Computing (IDC 2021).

This paper is an extension and improvement of the given work. The model of microcontroller-based systems was reconsidered and improved. First, the model of links between building blocks, software and hardware elements was extended with information about communication parties. Secondly, the model of the attacker now allows to distinguish attackers according to their type of access, knowledge and resources instead of level of capabilities and type of access. Thirdly, the model of attack actions now distinguishes actions according to their class, object and subject instead of preconditions.

Moreover, classes of attack actions are specified in accordance with the levels of the system they occur in, while examples of actions of each class are provided. Finally, the connections between models of the attacker, attack actions, security and non-security elements are described in detail. Unlike before, the workflow of the methodology is described as a set of algorithms that are combined into a single automated approach with minimal operator involvement.

For the first time, the software implementation of the methodology is presented. It is used to validate the correctness of the methodology based on the design of a secure system of mobile robots for perimeter monitoring.

The methodology presented in this paper works only with ready-made components and controllers, without taking into account elements of electronic circuits. It is not generating source code of the system software and firmware. The parameters of the device case, its cooling and resistance to various weather conditions are not considered. This indicates that, for example, navigation of mobile robots is considered only on the level of required sensors and algorithms, while the process of constructing such algorithms is a separate complex task [25,26].

The paper is organized as follows. Section 2 considers the state of the art in the area of design of secure microcontroller-based systems. In Section 3 the original extendable set-based hierarchical relational model of microcontroller-based physical security systems is presented. Section 4 describes the new approach for the design of microcontroller-based physical security systems. In Section 5 an experimental evaluation of the developed methodology is presented. In Section 6 advantages and disadvantages of the methodology are considered. Section 7 contains general conclusions and future work directions.

## 2. Related Work

There are many approaches to ensure the information security of microcontroller-based systems [27]. As a rule, they are associated with individual stages of such systems development lifecycle: analysis, planning, design, development, testing, deployment, maintenance and evaluation; see Figure 1.

Approaches that are discussed in this section are used at the design stage of the systems development lifecycle and associated with the security by design approach. This is an approach to software and hardware development that aims to reduce the amount of possible vulnerabilities and enhance the system’s protection against possible attacks. The main idea of the approach is in taking into account security features as a design criterion of products.

Moreover, based on the information about the component composition of a system based on microcontrollers, it is possible to determine a list of attack actions to which this system is potentially susceptible. Then, based on the parameters of the attacker, this list of attacks can be limited in the same way as if there is information about the methods and means of protection used. All attack actions remaining are representing a real threat and must be considered. Information about attack actions to which the designed system is potentially susceptible is often used by design techniques to find a trade-off between the level of protection of the resulting solution and resources expended on it.

The task of designing microcontroller-based systems to be secure against attacks is complex, and thus different techniques in this area are widely used for specific areas of applications—robots, railway infrastructure, smart cities, etc. Those techniques can be aimed at software, hardware, individual devices and classes of systems. Let us consider them in more detail.

The design process of resilient microcontroller-based systems is presented in [13]. According to the authors, the resilient system includes three features: stability, security and systematicness. *Stability* means that the control system always reaches a stable decision result eventually. *Security* means that the designed system is able to detect and countermeasure cyber-physical attacks. *Systematicness* means that cyber and physical components are integrated together. The system framework suggested by the authors includes six levels: sensing, processing, modeling, decision fusion, human and actuators. To design a resilient microcontroller-based system, the following challenges should be addressed: reliability, dependability, consistency, cyber-physical mismatch and coupling security.

In [7], a microcontroller-based automatic power factor correction technique was implemented by the authors. The goal of the system is to minimize penalties, reduce losses and save power. The authors used a PIC18F452 microcontroller to build a system prototype. The design approach consists of (1) development of the block diagram of the entire system; (2) development of the circuit diagram of the system; (3) development of the system prototype; and (4) prototype testing and evaluation. All of the approach steps were done manually.

The extension of mechatronic systems to cyber-physical ones is presented in [14]. The authors considered similarities of such systems and underlined the need of cyber-physical systems in the manufacturing sector. Moreover, the authors presented the main research issues in the process of designing such systems, namely, the need for the integrated and multi-scale approach. Such an approach is required to prevent cross-domain conflicts in the process of development of cyber-physical systems.

The authors described the requirement for multi-scalability based on external and internal interactions, process control, behaviour simulation, topological relations and interoperability. The suggested design methodology contains seven main stages—namely, system boundary definition, multi-view/multi-level modeling, interactions modeling, topological modeling, semantic interoperability, multi-agent modeling and collaboration modeling. As a case study, the authors used a process of manufacturing tablets.

A microcontroller-based wireless humidity monitor was designed and developed in [8]. The authors used a DHT22 sensor module and an RF 433 MHz transmitter that were connected with an Arduino Uno microcontroller. The collected data was displayed on a 16 × 1 LCD. The author’s approach to design such a system contains (1) design of the block diagram, which divided the system into the transmission and receiving sections; (2) design of flow charts for each section of the system; (3) development of the software for each part of the system; and (4) implementation of the system. The authors built such a system manually, using their own knowledge.

In [9], the authors identified the requirements for a secure, robust and resilient SDN (Software-Defined Network, [28]) controller. Moreover, such controllers were analyzed with respect to the security of their design, and recommendations for security improvements were provided. The security attributes were categorized into three groups: secure controller design, secure controller interfaces and controller security services. As for design, the following functionality was checked: control process (application) isolation, implementation of policy conflict resolution, multiple Controller instances, multiple application instances and secure storage.

A systematic methodology for a chaotic map-based real-time video encryption and decryption system was proposed in [10]. Moreover, the authors developed its hardware implementation and tested it in a real-world network environment. The security performance of the designed system was tested using criteria from the National Institute of Standards and Technology statistical test suite. Based on theoretical analysis and experimental results, the authors concluded validity and feasibility of the new secure video communication system.

Google Cloud Internet of Things is a set of technologies that is focused on network-connected devices that are embedded in the physical environment [16]. As an additional dimension of microcontroller-based systems compared to other cloud applications, the authors highlighted the following: diverse hardware, operating systems and software on devices as well as requirements for network gateways. This solution divides microcontroller-based systems into devices (software + hardware), gateways and clouds.

The main benefit of this technology is in the possibility to work with Google services, including artificial intelligence ones. For example, Vertex AI can be used to work with machine learning tools developed by Google Research [29]. To connect microcontroller-based devices to Google cloud, the Google IoT Device SDK (software development kit) was developed [30]. This SDK supports a wide range of 32-bit microcontrollers and real-time operating systems (Zephyr, ARM Mbed OS, FreeRTOS, etc.), while libraries of SDK are written in Embedded C and available on Github [31].

PSA Certified is a security framework for the Internet of Things that was created by ARM, Brightsight, CAICT, Prove & Run, Riscure, TrustCB and ULs [17]. This framework is an extension of the ARM Platform Security Architecture (PSA). The idea of the framework is in providing an approach for security design of connected microcontroller-based devices. The main features are as follows: open resources, layered approach, reduced costs, standardized security, unbiased certification and aligned standards. As the main 10 security goals the following are mentioned: unique identification, security lifecycle, attestation, secure boot, secure update, anti-rollback, isolation, interaction across isolated boundaries, secure storage and cryptographic and trusted services.

For additional security on a hardware level, the authors developed a PSA Root of Trust (PSA-RoT), which is the the part of the chip that performs trusted functions to ensure security [32]. The PSA Certified framework also provides a security-by-design methodology that consists of four main steps: analysis and identification of security requirements; consideration of the security architecture; integration of the components; security assessment and certification [33].

Kaspersky Industrial CyberSecurity (KICS) is a combination of products and services for industrial-level cybersecurity [18]. It contains the following products: KICS for nodes (endpoint protection), KICS for networks (anomaly and breach detection) and Kaspersky Security Center (security management). Services are divided into training and awareness (Kaspersky security awareness and cybersecurity training) as well as expert services and intelligence (Kaspersky security assessment, incident response and threat intelligence).

As benefits, the following are mentioned by the authors: asset discovery, deep packet inspection, network integrity control, intrusion detection, command control, anomaly detection, vulnerability management, external systems integration, reporting and notification. The design of secure industrial microcontroller-based devices is based on Kaspersky Operation System (KasperskyOS) [34].

Technologies of KasperskyOS include cyber immunity (sensitive components isolation), microkernel (links between software and hardware) and security policies configuration (policies customization, internal communications scanning). The work process of this operating system is based on prohibition of all actions recognized as dangerous.

The main technologies of Microsoft for Internet of Things are Azure IoT Hub [19] and Digital Twins [35]. Azure IoT Hub provides a possibility to design secure and reliable communication between microcontroller-based devices and cloud applications. The main features of this technology are as follows: per-device authentication, built-in device management and scaled provisioning.

Security risks are reduced with the help of Azure Defender for IoT [36]. Azure Digital Twins is a platform for modeling devices, places, business processes and people. As a modeling language, Digital Twin Definition Language (DTDL) is used, its specification is available on the Github repository [37]. In addition, all Microsoft solutions are developed with the help of their Security Development Lifecycle (SDL) [38].

This solution consists of the following stages: personal training, security requirements definition, acceptable levels of security definition, vulnerabilities identification, risk and countermeasures analysis, security features definition for the design, cryptographic solutions definition, third-party components analysis, approvement of tools, static and dynamic analysis security testing, penetration testing, incident response plan preparation. As a benefit, Microsoft SDL helps developers to build software with reduced number of vulnerabilities, while the development process becomes cheaper.

Intel Internet of Things is a platform for the secure connection of microcontroller-based devices to the cloud via gateways [20]. This platform ensures that data is efficiently collected, processed and delivered. Moreover, this platform is integrated with the IBM Watson IoT platform [39]. The IBM platform is a cloud service that provides devices and their data management functions. The development of secure microcontroller-based devices is based on the Intel oneAPI IoT Toolkit [40].

This toolkit helps with system design, development and deployment across CPU, GPU, FPGA, SoC and other architectures. The key benefits of FPGAs and SoCs are as follows: customization of both hardware and software; security through hardware cryptography; secure in-field upgrades [41].

MindSphere is a cloud-based service solution for Industrial Internet of Things from Siemens [21]. It is used for contextualized collection, analysis and visualisation of data. The key features of the solution are as follows: advanced analytics of near-real-time data; connection of physical and web systems in one solution; integration of industrial solutions; utilization of multiple data services. Analytics are based on artificial and business intelligence, machine learning, industrial edge and visualisation [42].

Connectivity is based on abstract representation of devices, systems and workflows, support of a variety of protocols, availability of out-of-the-box solutions as well as possibility of integration with enterprise IT systems [43]. Integration is based on closed-loop digital twin, condition monitoring, low-code application development and modular offerings [44]. The connection of the application to the MindSphere is based on MindConnect API [45]. This API provides microcontroller-based devices with a possibility to send data to the MindSphere service securely and reliably.

A configuration model for the design of microcontroller-based devices that are secure and efficient in terms of the resource consumption is presented in [12]. This model is used for the search of rational combinations of security building blocks based on their resource consumption.

The authors proposed an approach for determining the process of combination of individual algorithms and techniques that are implementing different protection functions. Moreover, it was noted that the developed model and approach should be used as a part of the design process for the whole microcontroller-based system. As a basis for the microcontroller-based devices analysis, MARTE is used [46]. MARTE is a UML profile for the modeling and analysis of real-time embedded systems.

The configuration model consists of: raw data (device specification, description of security building blocks), input data that is extracted from the raw data (functional and non-functional properties, attacker and threat models, security requirements, resources and compatibility of the devices) as well as functions of the configuration process (filtration, verification, compatibility analysis and optimization-based composition). The model was applied for the smartphone HTC Wildfire S with Android 3.2.

In [11] a technique for the design of secure and energy-efficient microcontroller-based devices was presented. This technique finds out rational combinations of security components on the basis of solving the optimization problem. According to the authors, devices are designed in accordance with the structured sequence of actions. Security components are combined using semantic rules for their choice in accordance with functional and non-functional constraints.

The main stages of the technique are as follows: definition of functional and non-functional constraints; identification of security components in accordance with functional constraints; definition of rules for the selection of security components in accordance with the relations; calculation of non-functional constraints; choice of the rational configuration in accordance with the importance of non-functional constraints. This technique was applied for a perimeter protection system in its part of the implementation of the room access control.

A methodology for the network information flow analysis in microcontroller-based systems was presented in [47]. According to the authors, the main security issue of such systems lies in the fact that they work in a potentially hostile environment, while having strong resource limitations. The following levels of the systems were mentioned and applicable for the security-critical information flow analysis: electronic circuit schemes (hardware flows), data of devices firmware and software (software flows), and communications between devices (network flows).

The suggested methodology consists of the two main approaches: topological and security policies based information flow analysis. Topological approach is based on the identification of all components that are lying between two nodes of the graph, between secure information source and not secure information target. The security policies approach uses filtering rules, network configuration and a description of anomalies for the detection of conflicts in policies with the help of model checking tools. This methodology was developed in the framework of the SecFutur project [15].

The main disadvantages of the analyzed scientific solutions are as follows:a focus on only one aspect of the security of microcontroller-based systems;strong relations between software and hardware in microcontroller-based devices are not considered;microcontroller-based devices are designed without taking into account the system they will work in; andsecure communication is provided only between microcontroller-based devices, while their communications with external systems is not considered.

The main disadvantages of the analyzed commercial solutions are as follows:strong restrictions on the possible platforms and architectures of microcontroller-based devices to be used;security is ensured only on the level of cloud services and communications between the gateway device and the cloud; andthere is no trade-off between the security level and resources expended on it.

This means that a general approach for solving the issue of secure microcontroller-based systems design is not done yet. Therefore, an original approach for the design of secure microcontroller-based systems is required. Such an approach should work with an abstract representation of the designed system, find a trade-off between the expended resources and the obtained level of security, have no restrictions on the platforms and architectures of the designed devices and take into account the physical layer of the designed systems. Moreover, the new approach should be extensible and modular and have a strong focus on security.

## 3. Hierarchical Model

To display various aspects of complex systems and detect the potential feasibility of various attack actions component-based, semi-natural, simulation and analytical modeling are used. Each modeling approach has its own abstraction level in the representation of the system [48]; see Figure 2.

The component-based approach is the most detailed way to represent microcontroller-based physical security systems; however, it requires a great deal of time and effort. Moreover, it is not possible to represent different dynamic processes with it. From the other side, with the help of analytical modeling, it is possible to represent the whole system but only on a high level of abstraction. Therefore, the performance of the solution strongly depends on the level of detailing. That is why, to represent the whole lifecycle of the system, heterogeneous structures of the united models are used to overcome this issue by using different models for different cases.

For the design process of microcontroller-based physical security systems the component-based approach is the most appropriate one if it is required to take into account the security of the system as early as possible. The model suggested in this work represents such systems as an extendable set-based hierarchical relational structure and consists of the following parts: building blocks (hardware and software elements), links between system elements (protocols and interfaces) and an attacker and attack actions.

One of possible ways to describe complex systems as a set of interacting building blocks is the set-theoretic approach. Let us consider it in more detail.

### 3.1. Microcontroller-Based Physical Security Systems

Any system mbs∈MBS can be represented as follows:(1)$$mbs=(MBS^{{}^{\prime }},BB,L_{mbs},a,AA,p_{mbs}),$$
where $MBS^{{}^{\prime }}$ is the set of microcontroller-based sub-systems of mbs; BB is the set of building blocks of mbs; $L_{mbs}$ is the set of links between BB and $MBS^{{}^{\prime }}$ of mbs; *a* is the attacker against mbs; AA is the set of attack actions on mbs; and $p_{mbs}$ represents the properties of mbs.

It is important to note that each element of the system at this level is considered as an object with a set of properties and links without taking into account its internal structure. This rule is functional for the sub-elements of each element as well.

The model of the system allows one to represent the information about its sub-systems through $MBS^{{}^{\prime }}$ and its individual blocks through BB. Information about the data transfer environment between sub-systems $mbs_{i}^{{}^{\prime }}\in MBS^{{}^{\prime }}$ and individual blocks $bb_{j}\in BB$ is represented through $L_{mbs}$, while properties arising from their interaction are represented through $p_{mbs}$.

As an example of mbs, any microcontroller-based physical security system can be used: a naccess control system, fire alarm system, security alarm system, closed-circuit television system, perimeter monitoring system, etc. The situation when mbs contains sub-systems is related to integrated physical security systems that combine, for example, access control and fire and security alarm systems.

A building block of mbs can be represented as follows:(2)$$bb=(BB^{{}^{\prime }},HW,SW,L_{bb},p_{bb}),$$
where $BB^{{}^{\prime }}$ is the set of building sub-blocks of bb; HW is the set of hardware elements of bb; SW is the set of software elements of bb; $L_{bb}$ represents the links between elements of bb; and $p_{bb}$ represents the properties of bb.

The model of individual blocks bb∈BB allows one to represent the information about its sub-blocks through $BB^{{}^{\prime }}$, hardware through HW and software through SW. Information about the data transfer environment between individual sub-blocks $bb_{i}^{{}^{\prime }}\in BB^{{}^{\prime }}$, hardware $hw_{j}\in HW$ and software $sw_{k}\in SW$ is represented through $L_{bb}$, while properties arising from their interaction are represented through $p_{bb}$.

As an example of a building block, any device, controller or its combination with components can be used. For example, it can be a Raspberry Pi single board computer, micro SD card with a pre-installed operating system, ESP8266 or Iskra JS microcontroller or even a server, hub, robot, station, drone, etc.

A hardware element of mbs can be represented as follows:(3)$$hw=(HW^{{}^{\prime }},L_{hw},p_{hw}),$$
where $HW^{{}^{\prime }}$ is the set of hardware sub-elements $hw^{{}^{\prime }}$ of hw; $L_{hw}$ represents the links between elements of hw; and $p_{hw}$ represents the properties of hw.

The model of individual hardware elements hw∈HW allows one to represent the information about its sub-elements (also hardware) through $HW^{{}^{\prime }}$. Information about the data transfer environment between individual sub-elements $hw_{i}^{{}^{\prime }}\in HW^{{}^{\prime }}$ is represented through $L_{hw}$, while properties arising from their interaction are represented through $p_{hw}$.

As an example of a hardware element, any component can be used: sensors, receivers, transmitters, readers, motors, batteries, etc. As an example of a hardware element that consists of multiple hardware elements, let us consider a motor shield with two collector motors that can be used for two-wheel robots. When motors are connected to the motor shield, their rotation speed and direction are controlled by its signals, while the controller of the robot can be connected to the motor shield to control signals of the shield through the firmware.

A software element of mbs can be represented as follows:(4)$$sw=(SW^{{}^{\prime }},L_{sw},p_{sw}),$$
where $SW^{{}^{\prime }}$ is the set of software sub-elements of sw; $L_{sw}$ represents the links between elements of sw; and $p_{sw}$ represents the properties of sw.

The model of individual software elements sw∈SW allows one to represent the information about its sub-elements (also software) through $SW^{{}^{\prime }}$. Information about the data transfer environment between individual sub-elements $sw_{i}^{{}^{\prime }}\in SW^{{}^{\prime }}$ is represented through $L_{sw}$, while properties arising from their interaction are represented through $p_{sw}$.

As an example of a software element, any algorithm, library, firmware, database, application or configuration can be used. As an example of a software element that consists of multiple software elements, let us consider a firmware of the controller that can be used as the brain of a two-wheel robot. Such a firmware often contains library imports for most components that are connected to the controller as well as algorithms for the correct functioning of the robot: navigation, communication, data processing and storage etc.

Links between elements of mbs can be represented as follows:(5)$$L=(R,I,E,p_{L}),$$
where *R* is the set of protocols that are used in *L*; *I* is the set of interfaces that are used in *L*; *E* is the set of communication parties of *L*; and $p_{L}$ represents the properties of *L*.

The model of individual links l∈L allows one to represent the information about its protocols through *R*, interfaces through *I* and communication parties through *E*, while properties arising based on their combination are represented through $p_{L}$.

Moreover, links between elements of mbs can be divided:$L_{mbs}$ represents the links between devices of the system.$L_{bb}$ represents the links between controllers of devices.$L_{hw}$ represents the links between controllers and components.$L_{sw}$ represents the links between software elements.

This indicates that the model allows one to represent low level protocols between controllers and components together with connections between different algorithms inside the firmware of one of controllers, while being able to represent high level protocols between devices; see Table 1.

Within the framework of the developed model, all elements are connected with each other through their properties. Thus, to ensure the required level of security of the designed system, the goal of the approach is to find a reasonable combination of elements of the system according to the balance between their needs (functional requirements and non-functional limitations) and capabilities (provided functionality and resources). On the other hand, the influence of each successful attack action is represented through a reduction of the system capabilities (for example, denial of service) or enhancing of its needs (for example, resource depletion).

Thus, the properties can be represented as follows:(6)p=(FR,NL,PF,PR),
where FR is the set of functional requirements (the functionality that satisfaction is necessary for the element to work); NL is the set of non-functional limitations (the limitation that satisfaction is necessary for the element to work); PF is the set of provided functionalities; and PR is the set of provided resources.

The model of properties p allows one to represent the information about elements needs and capabilities through FR, NL and PF, PR accordingly. Let us consider examples of each of them in more detail.

As functional requirements of the element, any functionality necessary to be able to work can be used—a power source, secure connection, protocol, interface, bootloader, library, operating system, compiler, driver, etc.

As non-functional limitations of the element, any limitation necessary for it to be able to work can be used—a space for placement, suitable environment, voltage, current, size, volume, flash memory, digital or analog pins, disk space, ram, etc.

As provided functionality of the element, any functionality that it can provide can be used—access control, perimeter monitoring, navigation, obstacles detection, work with a component, encryption, authentication, processing, etc.

As provided resources of the element, any resource it can provide can be used—data storage, computing resources, environment for launching applications, the possibility to add/remove/replace components, the possibility to work with environment, etc.

### 3.2. Attackers, Attack Actions and Security Elements

An attacker against mbs can be represented as follows:(7)a=(ac,kn,rs),
where ac is the type of access *a* has to mbs; kn is the type of knowledge *a* has about mbs; and rs is the type of resources available to *a* to compromise mbs.

According to the developed model, the attacker’s ac can be in a range between 1 and 5. This value describes the type of access an attacker has with the microcontroller-based physical security system; see Table 2.

An attacker’s kn can be in a range between 1 and 4. This value describes the amount of information available to the attacker about the system; see Table 3.

An attacker’s rs can be in a range between 1 and 3. This value describes the amount of resources available to the attacker; see Table 4.

In the developed model, the structure of the attacker’s access, knowledge and resources types is hierarchical. This indicates that $a_{1}$ with $ac_{a_{1}}=3$ is able to perform any attack action that is possible for $a_{2}$ with $ac_{a_{2}}=2$ if $kn_{a_{1}}\geq kn_{a_{2}}$ and $rs_{a_{1}}\geq rs_{a_{2}}$. It also means that $a_{3}$ with $ac_{a_{3}}=3$ is able to perform any attack action that is possible for $a_{1}$ if $kn_{a_{3}}\geq kn_{a_{1}}$ and $rs_{a_{3}}\geq rs_{a_{1}}$. However, if there are $a_{4}=\left(ac_{a_{4}}=3,kn_{a_{4}}=2,rs_{a_{4}}=2\right)$ and $a_{5}=\left(ac_{a_{5}}=2,kn_{a_{5}}=3,rs_{a_{5}}=3\right)$ then $a_{4}$ will not be able to perform all attack actions that are possible for $a_{5}$ and vice versa.

An attack action on mbs can be represented as follows:(8)aa=(cl,oj,sj),
where cl is the class of aa; oj is the object of aa, which helps to link aa with the target element(s) of mbs; and sj is the subject of aa, which helps to link aa with *a*, which is sufficient for its successful realization.

In this work, instead of separate impact methods, we decided to use classes of attack actions, where each class contains multiple examples of methods.

Classes of attack actions can be represented as follows:(9)cl={cn,cr,dv,st},
where cn is the aa on the level of components and their communications with controllers they are connected to; cr is the aa on the level of controllers and their communications with other controllers; dv is the aa on the level of devices and their communications with other devices; and st is the aa on the level of the system and its communications with other systems.

Examples of attack actions on the cn level can be represented as follows:(10)cn={gie,bcd,rpt,rmt},
where gie is the generation of incorrect component events; bcd is the bypassing component detection algorithms; rpt is the replacement of the component; and rmt is the removal of the component.

Examples of attack actions on the cr level can be represented as follows:(11)cr={rfw,rbl,mup,imw},
where rfw is the replacement of the controller’s firmware; rbl is the reinstallation of the controller’s bootloader; mup is the malfunction of the controller’s update system; and imw is the interception, modification or termination of wired communications.

Examples of attack actions on the dv level can be represented as follows:(12)dv={vau,cad,iec,iws},
where vau is the violation of the authentication system; cad is the cryptographic analysis of transmitted data; iec is the increased energy consumption; and iws is the interception, modification or termination of wireless communications.

Examples of attack actions on the st level can be represented as follows:(13)st={soc,pwr,web,dbd},
where soc is the social engineering; pwr is the power failure; web is the disruption of web services; and dbd is the disruption of database services.

As individual security element, any means or method or protection can be used: an anomaly detection algorithm, hidden placement of sensors, events correlation algorithm, vandal-proof device case, hardware authentication, firmware encryption, bootloader encryption, removal of physical update interface, strong login credentials, password policy, brute-force protection, strong encryption algorithms, secure key distribution mechanism, behavior-based anomaly detection, devices isolation/limitation, training of operators and users, etc.

This indicates that most security elements can be modeled as software or hardware elements of the system and be integrated into its building blocks, while some of them can be transferred as recommendations to the designed system implementation.

### 3.3. Connections between Models

Let us consider how classes of attack actions are connected with the parameters of attackers; see Table 5. For example, let us consider the stakeholder that wants mbs to be secure against a=(ac=4,kn=2,rs=2). The gray coloring of the table cells represents values of the attacker’s parameters. Connections between the possibility to implement attack actions and values of the attacker’s parameters are shown with “+”. According to the content of the table, the designed system must be secure against: rpt, rmt, imw, iec, iws, soc, pwr, web and dbd. These attack actions are shown in purple.

As we mentioned before, the structure of attacker types is hierarchical. This indicates that an attacker with a certain access is able to perform any attack action that is possible for an attacker with the same access but with lower knowledge/resources. Such a dependence allows storing data only about the threshold values of the types that are necessary for the successful implementation of attack actions. It is important to note that the developed model allows the use of various models of attackers and attack actions. Thus, the number of attacker parameters, just like the permissible ranges of their values, can be changed. Likewise, for attack actions, another classification can be used, and the examples can be extended.

It is important to preserve the hierarchical nature of the attacker’s model and the relationship between the attacker’s parameters and the possibility of implementing attack actions. In addition, let us consider how classes of attack actions are connected with security elements of microcontroller-based systems; see Table 6.

In the developed model, possible attack actions are defined by the system element composition and parameters of the attacker, against which the system needs to be protected. This indicates that, if the possible attack actions are known, then the necessary security elements can be extracted. After that, each security element can be interpreted as software (for example, an anomaly detection algorithm), hardware (for example, a vandal-proof device case) and recommendations (for example, the training of operators and users).

Let us consider how classes of attack actions are connected with non-security elements of microcontroller-based systems; see Table 7. Relations between attack actions and non-security elements define the attack surface of the system. Understanding the attack surface allows its reduction in early stages of the system life cycle, significantly increasing its security level.

## 4. Design Approach

The design approach presented in this paper is based on the developed methodology for the design of microcontroller-based physical security systems. The main idea of the methodology is to provide an automated tool for the design of microcontroller-based physical security systems that are protected against attackers. This methodology allows one to reduce the amount of weak places and architectural defects, thereby, significantly reducing the attack surface of the designed systems. In turn, this reduces the security risks that can lead to financial loss, loss of time as well as the safety of people.

The work process of the methodology is mostly automated, and involvement of the operator is required during the transformation of wishes of stakeholder into requirements and limitations and optional at the stage of selection of the concrete implementations of elements among suitable ones during the process of detailing the abstract system model. Alternatively, the methodology can select implementations on its own.

Let us consider the proposed methodology in more detail.

### 4.1. Proposed Methodology

The methodology for the design of microcontroller-based physical security systems protected from cyber-physical attacks consists of four algorithms; see Figure 3.

The goal of the first three algorithms is to design the abstract system model based on requirements, while the last algorithm design of the detailed system model is based on the available components. It is important to note that all algorithms of the methodology can be useful to an expert in the design of secure systems separately; however, their full potential is revealed only when they are interacting with each other.

The novelty of the methodology lies in a new approach to the design, which allows combining various design techniques on the basis of hierarchical relational model transformation algorithms [23]. The suggested approach is modular and extensible, takes into account the security of the physical layer of the system, works with the abstract system representation and is looking for a trade-off between the security of the final solution and expended resources.

Unlike existing solutions, the methodology has a strong focus on security. It aims at ensuring the protection of the system against attacks at the design stage, considers security components as an integral part of the system and checks if the system can be designed in accordance with given requirements and limitations.

Moreover, the methodology is not aimed to replace security experts. In most situations, an expert in the security of microcontroller-based systems knows about existing best and highly specialized solutions and is able to form alternatives at a very high level, while the quality of the solution provided by the methodology directly depends on the correctness and completeness of the database. However, it can be useful for an expert to automate routine tasks and provide alternative solutions.

Let us consider the algorithms of the methodology in more detail.

### 4.2. Algorithm for the Formation of Requirements to the System

This algorithm is used to extract attack actions that are possible for the attacker as well as a list of devices of the designed system, their links, communications, bases and requirements in accordance with the attacker’s parameters and system tasks.

As **input data**, the algorithm takes the:*attacker’s parameters*: characterizing the capabilities of the attacker in accordance with the developed model; and*system’s tasks*: characterizing the main tasks of the designed system in accordance with the wishes of the stakeholder.

As **output data**, the algorithm provides the:*attacker’s actions*: the list of attack actions that are possible for the attacker;*security elements*: the data structure for the security elements that are required to prevent possible attack actions is JSON-based;*devices list*: the list of devices that are required to design in accordance with the general tasks to the system;*device requirements*: the data structure for requirements for devices is JSON-based, while its keys are devices from the list of devices; by each device key, the data about requirements for this device can be extracted;*device communications*: the data structure for devices types of communications is also JSON-based, while its keys are also devices from the list of devices; by each device key, the data about possible for this device types of communications can be extracted;*device links*: the data structure for links between devices is JSON-based, while its keys are devices from the list of devices; by each device key, the data about its links with other devices can be extracted; and*device bases*: the data structure for bases of devices is JSON-based, while its keys are devices from the list of devices; bases are representing individual controllers or their combinations that are necessary for the device to work.

The work process of the algorithm is automated and contains six main stages, while the last stage is divided into seven sub-stages. The operator is required for the translation of wishes of the stakeholder into the attacker’s parameters and general tasks of the system. Let us consider each stage in more detail.

**Stage 1: Initialization of data structures**. This stage defines the data structures for storing devices and their requirements, communications, links and bases. Devices are stored as a list, while their requirements, communications, links and bases are stored as dictionaries—key-value structures.

**Stage 2: Extracting attack actions that are possible for the attacker**. At this stage, data about attack actions that are possible for the attacker in accordance with his or her parameters is extracted. Possible values of parameters are predefined by the model of the attacker. Concrete values of parameters are provided as input data and selected by the operator, while connections between parameters and attack actions are predefined in the database.

**Stage 3: Extracting security elements to prevent attack actions**. This stage is aimed at the extraction of security elements that are required to protect the system against attack actions. Actions that are possible for the attacker are provided by Stage 2. Security elements are extracted for each action separately and combined.

**Stage 4: Extracting abilities of the designed system**. At this stage, data about abilities that are expected from the designed system is extracted. Abilities are extracted in accordance with the tasks of the designed system that are provided as an input data. Abilities can be interpreted as something that the designed system must be able to do to solve tasks. For example, the task “static perimeter monitoring” can be connected with the following abilities: “to communicate with mobile robots of the system”, “to provide wireless charging”, “to monitor the perimeter nearby” and “to communicate with the server of the system”. Tasks are selected by the operator.

**Stage 5: Extracting the requirements of the designed system**. This stage is aimed at extraction of data about requirements for the designed system. The requirements are extracted in accordance with the abilities of the designed system that are provided by Stage 4. The requirements can be interpreted as something that is required for the designed system to have abilities. For example, the ability “to provide wireless charging” can be connected with the requirement “device that represents the charging stations of the system”.

**Stage 6 is called extracting the device data**. At this stage, data about devices are extracted based on the requirements of the system, which are provided by Stage 5. This is done for each requirement separately. Let us consider it in more detail.

**Stage 6.1: Extracting the device name**. This stage is aimed at extraction of data about the name of the device based on the provided requirement for the designed system. The name of the device is based on the requirement’s description. For example, the requirement “device that represents the charging stations of the system” is transformed into “charging station”. Such a transformation is possible because of the description format “device that represents the [device name] of the system”. Extracted names of devices are stored in the devices list.

**Stage 6.2: Extracting device tasks**. At this stage, data about tasks that are expected from devices of the system is extracted. Tasks of devices are extracted in accordance with requirements for the system that are provided by Stage 6.1. Those tasks can be interpreted as functionality that the designed device must have to fulfill system requirements. For example, the system requirement “device that represents the charging stations of the system” can be connected with the following tasks of the device: “work cycle support”, “interaction with intruders”, “interaction with mobile robots” and “interaction with the server”.

**Stage 6.3: Extracting device abilities**. This stage is aimed at extraction of data about abilities of the designed devices. Abilities are extracted in accordance with the tasks of the designed devices that are provided by Stage 6.2. Abilities can be interpreted as something that the designed devices must be able to do to solve their tasks. For example, the task “interaction with intruders” can be connected with abilities “to detect intruders” and “to chase intruders”.

**Stage 6.4: Extracting device requirements**. At this stage, data about requirements for the designed devices is extracted. The requirements are extracted in accordance with the abilities of the designed devices that are provided by Stage 6.3. The requirements can be interpreted as something that is required for devices to have their abilities. For example, the ability “to detect intruders” can be connected with the “motion sensor”, “servo drive”, “noise sensor” and “detection algorithm” requirements.

**Stage 6.5: Extracting the device base**. This stage is aimed at the extraction of data about the bases of the designed devices. The bases are extracted in accordance with the requirements for devices that are provided by Stage 6.4. The base of the device can be interpreted as something that represents its main computing unit. In this work, the base can have the following values: “single-board computer”, “connected microcontrollers” or “microcontroller”. Such values were selected to represent controllers of microcontroller-based devices. The extraction of the base for each device is based on all its requirements, where the necessary base is selected according to the principle of minimum allowable computing power.

**Stage 6.6: Extracting the device types of communications**. At this stage, data about types of communications that are possible for the designed devices is extracted. Types of communications are extracted in accordance with the bases of devices that are provided by Stage 6.5.

**Stage 6.7: Extracting the device links**. This stage is aimed at extraction of data about links between the designed devices. Links are extracted in accordance with the abilities of devices that are provided by Stage 6.3.

Note that during most of the stages the algorithm relies on the content of the database for making decisions. The output data of the algorithm is well-structured, while the algorithm takes into account dependencies between stakeholder’s wishes and system tasks, system tasks and system abilities, system abilities and system requirements, system requirements and device tasks, device tasks and device abilities, device abilities and device requirements.

### 4.3. Algorithm for the Formation of the System Components Composition

This algorithm is used to extract abstract elements and sub-elements of the system devices, security recommendations to the implementation of the system and its devices as well as abstract links between devices based on attack actions that are possible for the attacker, list of devices of the system, their bases, types of communications and links and requirements for them. It works with abstract elements, links and recommendations and represents the designed system component composition as multiple devices, each of which has multiple abstract elements, while each abstract element can have multiple abstract sub-elements. Wherein abstract elements and sub-elements represent controllers and components as well their software, including those that are related to security.

As **input data**, the algorithm takes:*devices list*: the list of devices that are required to design;*devices bases*: the data structure for bases of devices is JSON-based, while its keys are devices from the list of devices;*devices requirements*: the data structure for requirements for devices is JSON-based, while its keys are devices from the list of devices;*attacker’s actions*: the list of attack actions that are possible for the attacker; each attack action has an id, name and description;*devices communications*: the data structure for devices types of communications is also JSON-based, while its keys are also devices from the list of devices;*devices links*: the data structure for links between devices is JSON-based, while its keys are devices from the list of devices.

As **output data**, the algorithm provides:*abstract elements and sub-elements*: abstract component composition of the system devices, where abstract elements are extracted based on requirements for the device and possible attack actions and represent controllers, components, software and firmware, while abstract sub-elements are extracted based on abstract elements and represent algorithms, settings and requirements; the data structure for abstract elements and sub-elements is JSON-based, while its keys are devices from the list of devices;*security recommendations*: abstract security recommendations to the system implementation as a whole as well as for each of its devices separately that are extracted based on security elements and cannot be interpreted as abstract elements or sub-elements; the data structure for recommendations is also JSON-based, while it has keys for the system and all its devices; and*abstract links and abilities*: abstract types of communications that are possible between devices of the system with corresponding devices abilities that are related to their interaction; data structure for links is JSON-based, while its keys are devices from the list of devices; by each key the data about the respective links can be extracted.

The work process of the algorithm is automatic and contains **two main stages**, while the last stage is divided into **five sub-stages**. The operator is not required. Let us consider each stage in more detail.

**Stage 1: Initialization of data structures**. This stage defines the data structures for storing abstract elements and sub-elements of devices, security recommendations to the implementation of the system and its devices as well as abstract links between devices and abilities that define those links.

**Stage 2 is called extracting the component composition of devices**. At this stage, based on the provided as input data devices list, component composition of each device of the system is extracted. Let us consider it in more detail.

**Stage 2.1: Extracting abstract elements with their sub-elements**. This stage is aimed at the extraction of data about abstract elements of devices of the system as well as their sub-elements based on the provided requirements for devices and their bases. Elements are extracted recursively based on: the provided device base; provided requirements for the device; and already extracted elements.

**Stage 2.2: Extracting possible attack actions**. At this stage, data about attack actions that are possible for the designed devices in accordance with their types of communications and component compositions are extracted. Types of communications that are possible for the device are provided as input data. After attack actions that are possible based on component composition and communications of the device are extracted, they are compared with attack actions that are possible in accordance with parameters of the attacker. Intersection of these two sets of attack actions allows finding the set of attack actions that are possible on the designed device.

**Stage 2.3: Extracting additional abstract elements with their sub-elements**. This stage is aimed at extraction of data about additional elements and sub-elements of the device based on the provided attack actions. Additional elements are related to the means and methods of protection that are necessary to prevent attack actions. First, the list of required security elements is extracted. After that, abstract elements and sub-elements that are representing security elements are extracted. In the end, additional elements of the device are combined with the other elements that were extracted in Stage 2.1.

**Stage 2.4: Extracting security recommendations for implementation**. At this stage, data about security recommendations to the implementation of the system and its devices in accordance with the security elements of devices is extracted. Recommendations can be interpreted as security requirements that cannot be satisfied on the component level, which is why they can be satisfied only after implementation. For example, a recommendation to the system can be as follows: “to educate operators and users of the system about social engineering attacks”.

**Stage 2.5: Extracting links**. This stage is aimed at teh extraction of data about links between devices of the system based on the provided input data–device links. This stage is related to transformation of the input data into another data structure called abstract links and abilities. The new data structure is JSON-based, while keys are devices from the list of devices and values are links between devices.

Once again, during most of the stages the algorithm relies on the content of the database for making decisions. The output data of the algorithm is well-structured, while the algorithm takes into account the iterative retrieval process of abstract elements of devices together with their sub-elements. At the beginning, abstract elements and sub-elements are retrieved in accordance with bases of devices, then on the basis of their requirements, after that in accordance with the already extracted elements and sub-elements as well as required methods and means of protection.

### 4.4. Algorithm for the Design of the Abstract Model of the System

This algorithm is used to construct an abstract representation of a secure system based on its devices list, their abilities, elements and sub-elements as well as security recommendations. It represents the system as an abstract hierarchical model that takes into account the connections between system devices, their elemental composition, dependencies between device elements and the requirements for them.

As **input data**, the algorithm takes:*security recommendations*: abstract security recommendations to the system implementation as a whole as well as for each of its devices separately that are extracted based on security elements and cannot be interpreted as abstract elements or sub-elements;*abstract elements and sub-elements*: the abstract component composition of the system devices, where abstract elements represent controllers, components, software and firmware, while abstract sub-elements represent algorithms, settings and requirements;*abstract links and abilities*: abstract types of communications that are possible between devices of the system with corresponding devices abilities that are related to their interaction;*security elements*: abstract methods and means of protection that are required to make the designed system secure against attackers with certain parameters, interpretable as security recommendations, abstract elements and sub-elements.

As **output data**, the algorithm provides the *abstract system model* that contains abstract system representation. The structure of the abstract model of the system is JSON-based and contains the following fields:*devices*: data about each device of the system, including its unique key, id, name, components and recommendations;*recommendations*: data about recommendations to the implementation of the system to ensure its security against attackers with certain parameters, including unique key, id and name (description);*links*: data about links between devices of the system, including the unique key, id, type, parties, dependencies and requirements.

Each element from the “components” field has its unique key and id as well as data about its own components (sub-elements), links, requirements and dependencies.

The work process of the algorithm is automatic and contains **seven main stages**. The operator is not required. Let us consider each stage in more detail.

**Stage 1: Initialization of the abstract model**. This stage defines the data structure for storing the abstract model of the system. At the end of the stage, the abstract model consists of fields for data about devices, links between them and security recommendations for the implementation of the system.

**Stage 2: Generation of the system security recommendations**. At this stage, the abstract model of the system is filled with data on the recommendations for the implementation of the system related to ensuring its security. Each of the recommendations has a unique key by which its id and text description are available.

**Stage 3: Generation of the system devices**. This stage is aimed at filling the abstract model of the system with data about its devices. For each device, data is generated about its unique identifier, name and component composition. Data on recommendations related to ensuring the security of devices after their implementation are also generated.

The main part of this stage is the generation of the device component composition. This part contains the initialization of abstract components of each device as well as the generation of their requirements based on each component sub-elements (including security ones). For example, depending on the component of the device, it is assumed how much flash memory of the firmware it needs to work correctly.

After this stage is done, each device of the abstract system model is filled with a number of elements in their “components” field. Each element represents an abstract component of the microcontroller-based system (the operating system, firmware, sensor, receiver, transmitter, database, microcontroller, etc). Each element in the abstract model has its own key that is unique only inside each device. By using this key, the data about its unique identifier, name, components, links and requirements can be extracted. It is important to note that data about each element’s components and links during this stage is empty and would be filled only during Stage 7.

**Stage 4: Generation of links between devices**. At this stage, the abstract model is filled with data on links between devices of the system. First, the algorithm detects all links that are possible between each pair of devices according to their abilities. If the link is detected, its generation starts. In the abstract model, each link has its own unique key, by which data about its unique identifier, type, parties, dependencies and requirements can be extracted. For example, the “dependencies” field is filled with data about abstract elements, the selection of a specific implementation of which directly depends on the selection of the interface and protocol of this link between devices. As an output of this stage, unique keys of links with unique identifiers of elements the selection of which depends on the selection of a specific interface and protocol of the link are provided.

**Stage 5: Generation of requirements for links**. This stage is aimed at filling the abstract model with data about requirements for links between devices. This field was empty after stage 4 and now is filled with data generated based on the information about security elements that are required to design a secure system. Generated during this stage requirements define if a link is wired or wireless, transfers data, signal or charge, requires encryption and/or authentication, etc.

**Stage 6: Generation of dependencies between elements**. At this stage, the abstract model is filled with data on requirements for elements of devices as well as with data about dependencies between them. For example, for each microcontroller data about dependencies between their selection and the subsequent selection of sensors that will be connected to them would be generated. It is done to ensure the compatibility of the elements of the device after the transmission from the abstract model to the implementation of the system. For each controller that is related to work with other components, like sensors, receivers and transmitters, the number of required digital and analogue pins is calculated.

**Stage 7: Generation of the hierarchy of elements**. This stage is aimed at the reconstruction of the “components” field of each device of the system. The algorithm generates a hierarchical element composition instead of their enumeration. The transmission to the hierarchical structure is based on a graph representation of the components of each device of the system and recursive conversions. First, graph nodes are generated based on unique identifiers and keys of elements. After that, elements of each device are checked pair by pair in terms of the possibility to connect one element to another. For example, a sensor can be connected to a controller if they are compatible, while compatibility can be checked according to their parameters. If two elements can be connected to each other then the edge between nodes that are representing them is generated.

After the graph structure for each device is generated, the process of hierarchy building starts. First, the root node of the graph is obtained based on topological sorting. After that, the child node of the lowest level of the graph is obtained together with its parent node. It is required for the algorithm to encapsulate the data about the obtained child element into the “components” field of its parent element as well as for the generation of a link between them. After it is done, the data about the encapsulated child is deleted from the abstract model (these data are in the “components” field of its parent now) and the node corresponding to this child is deleted from the graph representation of the device. This process continues until no other graph node can be deleted.

It is important to note that during Stages 3, 4, 6 and 7, the algorithm relies on the database for making decisions. The output data of the algorithm is well-structured, while the algorithm takes into account the hierarchy of elements and dependencies between them and generates requirements for them.

### 4.5. Algorithm for the Design of the Detailed Model of the System

This algorithm is used to construct a detailed representation of a secure system based on its abstract representation. The detailed model of the system preserves and expands the structure of the abstract model of the system and takes into account compatibility, requirements, dependencies and hierarchy of system elements. The process of transition from the abstract system model to a detailed one is a step-by-step process. Each step represents the process of selection of the concrete implementation of one of the system elements, while the sequence of steps is formed in accordance with hierarchy and dependencies between those elements. Moreover, after each step, the amount of options for further steps is limited in accordance with compatibility.

As **input data**, the algorithm takes the *abstract system model*.

As **output data**, the algorithm provides the *detailed system model*. The structure of the detailed model of the system is also JSON-based. Moreover, it has the same structure as the abstract system model but with some additions:each *element* from the *components* field that was selected is extended with the *selected* field: data about selected elements, including the id, name and parameters of its implementation; parameters of the element differ for different components and controllers;each *device* of the system is extended with the *parameters* field: data about the parameters of the designed device, including the price, energy consumption, voltage, current, length, width, height, free memory and battery life; device parameters are based on parameters of its elements; the parameters are mostly the same for all devices; however, the units for free memory are different for single-board computers and microcontrollers;each *link* between devices of the system is extended with the *selected* field: data about the selected links between devices, including id, name, interface, protocol and parameters; the parameters are the same for each link and can be divided into Boolean and numerical ones; Boolean parameters define if the selected link is wireless, directed, transfers data, charge or signal, requires an access point, and, if it has encryption or authentication; numerical parameters define the range and speed of the link.

The work process of the algorithm is automated and contains **six main stages**. Involvement of the operator is possible during the selection of concrete implementations of elements among suitable options. Alternatively, the algorithm can select them on its own. Let us consider each stage in more detail.

**Stage 1: Initialization of the data structures**. This stage defines the data structures for storing the selection steps and selected options. There is no need to define the data structure for the detailed model of the system, because it is stored in the same data structure that was used for the abstract model of the system.

The data structure for the selection steps is JSON-based and contains unique keys for each step of selection. Using this key, data about the selected element can be extracted. Each selected element has key, type, id, name, label, hierarchy, dependencies and requirements. There is also an additional field “selected” to store data about the selected options as well as the field “same for” that prevents the selection of one element multiple times.

The data structure for selected options is JSON-based and contains keys table and database id. By the table key, it is possible to extract data about the database table, where data on the selected option is stored, while database id is identificator of the concrete data tuple in the database table.

**Stage 2: Generation of selection steps based on links between devices**. At this stage, the sequence of selection steps is filled with data about selection of links between devices of the system. The sequence of selection steps is a very important part of the algorithm because of dependencies between components of devices as well as the possibility of their conflicts in terms of compatibility. That is why the generation of selection steps starts with selection of links between devices. Each link, after its selection, is limiting options for controllers and components that are related to communications between devices for compatibility.

**Stage 3: Generation of selection steps based on components of devices**. This stage is aimed at filling the sequence of selection steps with data about components of devices. This process is more complicated because of the hierarchical nature of device component compositions in the abstract model. In addition, it is important to take into account that components of one device can depend on the selection of components of another device. That is why first devices are selected in some order too, while data about each device component’s composition is extracted recursively. Moreover, the sequence of extracted components is also based on their hierarchy. Each element, after its selection, is limiting options for its dependable elements. For example, selection of the controller is limited to options for components that are connected to it for compatibility.

**Stage 4: Saving the data of selected options**. At this stage, the process of selection of concrete implementations begins. Each selection step means the choice of one option among suggestions. This process can be manually done by the operator or automatically by the algorithm. After the option is selected, the choice is saved so that it can be considered during the selection of other elements that have dependencies with the selected one. For example, if the link responsible for communication between devices of the system represents a Wi-Fi connection, the options for controllers are limited to those ones that support Wi-Fi or can be extended to support it. The list of options is based on the content of the database, while it can be limited according to the requirements of the abstract representation of the selected element. For example, requirements for the controller can limit its options to those that have at least the necessary amount of flash memory and pins. Thus, during this stage, all options that are representing the abstract element are limited in accordance with the compatibility, requirements and dependencies.

**Stage 5: Detailing of the abstract system model**. This stage is aimed at filling the abstract system model with the data of selected implementations of its elements and represents the process of detailing. As we mentioned during output data description, each selected element is extended with the selected field. This extension is based on the content of the database, while the selected options data structure provides data on the table where content is stored as well as the id of its tuple. For example, an element with name “single-board compute” can have selected field with the following key-values: Raspberry Pi 4 Model B 2 GB, Broadcom BCM2711 1.5 GHz, Cortex A72 4-core 64-bit, 2 GB RAM, 5 V, 3 A, 85 × 56 × 17 mm, 69 euro and 540 mA. The situation for each selected link is the same. For example, the link related to Wi-Fi connection between devices can have selected fields with the following key-values: Wi-Fi IEEE 800.11 2.4 GHz WPA2-PSK, 40 m range and 20 Mbit/s.

**Stage 6: Calculation of the device parameters**. At this stage, the parameters of devices of the designed system are calculated. As was mentioned in the output data description, those calculations are based on the parameters of the elements of devices and are mostly the same for all devices. For example, the parameters of the device that is representing a server of the system can be as follows: 106 euro, 540 mAh, 5 V, 3 A, 85 × 153.5 × 44.5 mm, 29,400 MB of free memory and 37 h of battery life. Note that the parameters of the system as a whole are not calculated, because the necessary amount of its devices is not known by the algorithm and depends on the concrete implementation of the designed system.

It is important to note that the algorithm relies on the content of the database when extracting options that can be selected as well as when checking parameters of the selected options. This indicates that the correctness of its work strongly depends on the content of the database. Its output data is well-structured, while the algorithm takes into account compatibility, requirements, dependencies and hierarchy of elements.

## 5. Experimental Evaluation

This section describes the experimental evaluation of the methodology for the design of microcontroller-based physical security systems. It contains the description of the designed system, software implementation of the methodology and the conducted experiment and its result analysis, including a comparison with analogues.

### 5.1. Use Case

We decided to design a microcontroller-based physical security system that provides perimeter monitoring based on mobile robots; see Figure 4.

This system contains a server as well as multiple mobile robots and charging stations with different controllers and components. Robots and stations monitor the perimeter via different sensors based on the server instructions. If the battery of one of the robots is low, it moves to the nearest free charging station. The information about the perimeter map, locations of robots and stations as well as the charge state of robots and occupancy of stations is stored on the server.

Such a system was chosen due to the presence of several types of devices, multiple communications between them, as well as the need to use many different elements for each device in the system (server consists of eight elements with sub-elements, station 12 and robot 17, which means that such a system is appropriate in accordance with the provided requirements).

Moreover, there are links between devices of the system and elements of devices, requirements for links and elements as well as dependencies between them. During the design of such a system, it is necessary to not only ensure its functionality (perimeter monitoring) but also to ensure that the system is secure against attacks on it.

As mentioned in Section 4, the extraction of requirements for the designed system starts from its tasks that are formulated by the operator in accordance with the wishes of the stakeholder. Tasks of the perimeter monitoring system are as follows:centralized system management;static perimeter monitoring;mobile perimeter monitoring; andan appropriate level of security.

Links between tasks, abilities and requirements are considered in Table 8.

Note that an appropriate level of security is set according to the parameters of the attacker model: access (ac∈Z,1≤ac≤5), knowledge (kn∈Z,1≤kn≤4) and resources (rs∈Z,1≤rs≤3) types.

Moreover, as noted in Table 8, the requirements can be divided into requirements for the server, mobile robots and charging stations of the system as well as security requirements. Note that security requirements should be considered not only on the system level but also during the design of all the devices.

#### 5.1.1. Server

Links between tasks, abilities and requirements are presented in Table 9.

Note that devices requirements are connected with controllers, components, etc.

#### 5.1.2. Charging Station

Links between tasks, abilities and requirements are presented in Table 10.

#### 5.1.3. Mobile Robot

Links between tasks, abilities and requirements are presented in Table 11.

It is important to note that devices of the designed system have requirements that introduce dependencies between their implementations:the *wireless network interface* must be satisfied for the server, charging stations and mobile robots in such a way that they can communicate with each other (the selected implementations must be compatible);the *wireless charge transmitters* of charging stations must be compatible withthe *wireless charge receivers* of mobile robots; andthe *wireless signal transmitters* of charging stations must be compatible with the *wireless signal receivers* of mobile robots.

The step-by-step design process of the microcontroller-based system is built inside the methodology in accordance with the relations between tasks, abilities and requirements. As a result of such a process, it is required to extract possible system component compositions that satisfy the provided requirements. Satisfaction of the requirements means that the designed system has abilities that are connected with those requirements. In turn, the abilities are connected with tasks, which means that the designed system is also able to solve the related tasks when the requirements are satisfied. If the system is able to solve the required tasks, then a correct microcontroller-based system was designed.

Note that each requirement could be linked to a number of element implementations. For example, a chassis of the robot can be with one or more wheels or tracks, and, depending on the selected option, an appropriate number of motors will be required for its movement. Moreover, some element implementations can be used to fulfill several requirements. For example, environment scanning sensors can be used to detect obstacles and/or intruders. In addition, implementations of algorithms may have their own requirements that vary from each other. For example, different environment data processing algorithms use data from different sensors; therefore, dependencies as well as conflicts between element implementations are possible and should be considered during the design process.

### 5.2. Software Prototype

Our software implementation of the methodology is an application that consists of Python script [49], PostgreSQL database [50] and Tkinter interface [51]. An overview of the software implementation architecture is presented in Figure 5.

The PostgreSQL database is required to store data about the extendable set-based hierarchical relational model of microcontroller-based physical security systems as well as data for algorithms and methodology. These data help to provide data to the operator as well as help the algorithms and methodology to make decisions about element compatibility, dependencies, hierarchy and nesting. The developed database contains more than 100 tables, while the database initialization contains more than 2300 lines of PL/pgSQL queries [52].

Python script represents the implementation of the algorithms and methodology. Each algorithm is implemented as a number of functions, while all functions are connected with each other in a single methodology. The developed script contains more than 3000 lines of code and works with imports, such as psycopg2 [53], tkinter [51], pygubu [54], networkx [55], json [56], functools [57] and time [58]. The role of the script is to implement algorithms, combine them together into the design methodology and provide connections between the database and the interface. The script connects itself with the developed interface based on the pygubu library. This allows the script to obtain access to objects of the interface and control them: the default state, selected values, callback functions and their links can be defined. The script connects itself with the developed database based on the psycopg2 library and its extension sql. This allows the script to extract data from the database.

The Tkinter interface is required to receive input data from the operator, namely, parameters of the attacker and tasks of the designed system as well as to provide the output data to him or her. The interface of the application consists of **six main parts**; see Figure 6:Input of the parameters of the attacker against which the designed microcontroller-based physical security system needs to be protected.Input of the tasks that need to be solved by the designed microcontroller-based physical security system.Frame to display the process of selection of components of the designed microcontroller-based physical security system.Frame to display the log of the work of the design methodology for microcontroller-based physical security systems.Frame to display the results of work of the design methodology for microcontroller-based physical security systems.Control buttons of the application.

Let us consider each part of the interface in more detail.

**Part 1**. Input of the parameters of the attacker is based on the model of the attacker and consists of three parameters: access type, knowledge type and resource type.

**Part 2**. Input of the tasks for the designed system is based on their selection from the developed database. For this demo, the number of possible tasks was limited to three.

**Part 3**. This frame displays options of communication protocols and interfaces, single-board computers, controllers and components to the operator. The choice made by the operator determines the detailed system model. It is important to note that the selection process is displayed step by step without the possibility of changing previously made decisions. Moreover, since each choice made affects the number of options available in subsequent steps, if there is only one option for selection, the choice is made automatically.

**Part 4**. This frame displays separate logs for the designed system, its devices, abstract and detailed models. *System log* contains information about attack actions that are possible for the selected attacker, security elements that should be used to prevent them, system abilities that were formed based on provided tasks, requirements that were formed based on these abilities and recommendations for the system implementation.

*Device logs* contain information about tasks that were formed for each device, abilities that were formed based on these tasks; requirements that were formed based on these abilities; the base of this device; its abstract elements, sub-elements and types of communication; attack actions that are possible based on types of communication, abstract elements and attacker parameters; security elements to prevent attack actions; additional elements of the device; additional sub-elements of the device; a generated set of device components and recommendations for the server implementation. *Abstract log* contains the abstract system model in JSON format, while *Detailed log* contains the detailed system model.

**Part 5**. This frame for each device displays the list of its components that were selected with their parameters as well as the list of components that are required to be developed or configured with required algorithms or settings. In addition, the parameters of each device as well as security recommendations for their implementation are displayed.

**Part 6**. Control buttons of the application are represented as “Design” and “Select” buttons and “automatic” checkbutton. *Design button* starts the design process for the abstract system model. *Select button* starts the selection process for the elements of the system to design its detailed model. *Automatic checkbutton* switches from the manual selection process (by the operator) to automated (by the methodology).

Using the developed interface, the operator can set the parameters of the attacker, against which the system is required to be protected, as well as tasks of the system. The interface is an important part of the software implementation, because it provides a possibility for the operator to work with the design methodology.

The source code of the script, the dump of the database as well as the file of graphical user interface are available for download using the following link: https://github.com/levshun/PhD-mcbpss_design (accessed on 5 December 2021).

### 5.3. Experimental Results

The model of the attacker characterizes the attacker’s capabilities in accordance with ac, which is the type of access; kn, which is the type of knowledge; and rs, which is the type of resources, where:ac can be in a range between 1 and 5 and describes the type of access that the attacker has to the system (for example, physical access to system devices);kn can be in a range between 1 and 4 and describes the amount of information available about the system (for example, system hardware and software is known); andrs can be in a range between 1 and 3 and describes the amount of resources available to the attacker (for example, the attacker can use specialized software tools).

The relationship between these parameters and dt, which is the system design time, is shown in Figure 7. The scale on the left from 0 to 5 reflects changes in values of the attacker’s parameters ac, kn and rs are shown as area charts, while the scale on the right from 0.27 to 0.35 reflects the design time; dt is shown as a black line.

The minimum design time was 0.2941 s, while the maximum was 0.3408.

To obtain the average design time for each parameter combination of the attacker, the software implementation was executed 100 times for each combination of values on the computer with Windows 10 × 64 operating system, Intel Core i7-4790 CPU 3.60 GHz (8 cores) processor, 2 TB HDD and 32 GB RAM. The time consumption was measured with the help of the time Python library.

It is important to note that, according to the related work analysis, there are no data available regarding the average time of design of microcontroller-based physical security systems by commercial or scientific solutions with which the developed one was compared. Moreover, even if these data were available, it is difficult to compare design approaches when different systems with different amounts of devices that contain different amounts of elements are designed.

### 5.4. Results Analysis

To evaluate the methodology for the design of microcontroller-based physical security systems, it is required to analyze the compliance of its software implementation with requirements as well as to compare the obtained results with scientific and commercial solutions.

According to Section 2, the requirements are as follows:Building an abstract representation of the designed system.Finding a trade-off between the resources spent and ensuring the security.No restrictions on platforms and architectures of the devices to be designed.The extensibility of the design process.Taking into account the physical layer of the designed systems.

Let us consider compliance with each requirement in more detail.

*Building an abstract representation of designed systems*. The abstract representation of the designed system is provided by the abstract model. This model is constructed by the algorithm that represents the system as an abstract hierarchical model that takes into account connections between system devices, their component composition, dependencies between device elements and requirements for them.

*Finding a trade-off between the resources spent and security*. The security of the designed system is based on the integration of security elements into its devices component composition. The amount of security elements that are required to be integrated depends on the number of classes of attack actions that are possible on the designed device in accordance with its component composition, communication levels and parameters of the attacker against which the system is required to be protected. This indicates that the exact same microcontroller-based physical security system can be designed with different amounts of security elements if parameters of the attacker differ.

*No restrictions on platforms and architectures of devices to be designed*. The software implementation first works with abstract elements, sub-elements and links and only after that replaces them with their implementations. This indicates that implementations of any platforms or architectures can be used in the developed solution, until they can be connected with abstract representations.

*The extensibility of the design process*. The structure of the database contains tables, the content of which affects how microcontroller-based physical systems are designed by the software implementation. Thus, the first way to extend the developed design approach is to fill those tables with more data: additional examples of attack actions, tasks, abilities, requirements, elements, sub-elements, etc. Moreover, it is possible to use other models of the attacker and attack actions. In addition, more parameters of elements can be considered as well as the list of calculated parameters for designed devices can be extended. Finally, additional algorithms can be integrated into the developed solution: the design of software, formal verification, solution of optimization problems, design on the level of electronic circuits, etc.

*Taking into account the physical layer of the designed systems*. The software implementation designs microcontroller-based physical security systems in accordance with the extendable set-based hierarchical relational model. This model represents systems, such as building blocks that are communicating with each other, while each block can have hardware and software elements. Moreover, communications between hardware and software elements are also considered as well as attack actions on them.

The results of analysis of commercial and scientific solutions, in accordance with the number of levels of the system, the security of which can be ensured and the number of classes of attacks against which the system can be protected, are presented in Table 12.

The levels of the system are divided into: cn↔cr, which is the controllers, components and their communications; cr↔cr, which is the controllers and their communications inside devices; dv↔dv, which is the devices and their communications with each other; and st↔st, which is the system and its communications with other systems.

Classes of attack actions are divided into: cn, which is the components and their communications with controllers; cr, which is the controllers and their communications with other controllers; dv, which is the devices and their communications with other devices; and st, which is the system and its communications with other systems.

The comparison showed that the developed design methodology provides protection against all analyzed classes of attack actions as well as takes into account security of all analyzed levels of microcontroller-based physical security systems, while other solutions do not consider that many parameters.

The main drawback of commercial solutions is that they are bound to the specific hardware, software, platforms and architectures. This indicates that, if the designed system already contains devices whose hardware cannot be changed or there are restrictions that do not allow the use of suitable devices, then these solutions are not applicable. In addition, these solutions do not take into account the optimization of the system design process due to limitations, such as the parameters of the attacker, computational complexity, energy efficiency and price. This indicates that resulting systems may not be reasonable for a developed use case as there is no trade-off between resources and the provided security level.

The main drawback of the existing scientific solutions is that they are focused on certain aspects of security, which ensures their inapplicability to provide the security of designed systems in general. For example, some approaches do not take into account that the functionality of system components is determined not only by software but also by hardware. Other approaches consider designed devices in isolation from the system they will work in.

This indicates that not all security aspects are considered and the security of the system as a whole will not be ensured. Some techniques are aiming at ensuring the security of communications between devices. The drawback is that such techniques provide secure connection between designed systems and external systems only from the designed side, which can lead to security issues during the design of complex multi-level systems.

It is assumed that the use of the developed solution will help to reduce the amount of weak places and architectural defects in microcontroller-based systems, thereby significantly reducing their attack surface. In turn, this will reduce the security risks that can lead to financial losses, loss of time as well as the safety of people, which ensures the relevance and high significance of this work.

## 6. Discussion

It is important to note that the developed solution is not aimed to replace experts in security of microcontroller-based systems. It is understandable that, in most situations, experts are aware of the existing best practices and highly specialized solutions and are able to design such systems at a very high level. However, even for an expert, it can be useful in terms of automating routine tasks as well as offering options that are different from those that are familiar.

The advantages of the developed solution are as follows:modularity and extensibility;the physical layer of microcontroller-based systems is considered;the designed system is represented in an abstract way first and then detailed;a trade-off between the security and resources expended is found; andsecurity components are considered as an integral part of the system.

The disadvantages of the developed solution are as follows:the quality depends on the correctness and completeness of the database;it is required to fill in the database manually; andthe selection of implementations is not optimized.

It is important to note that, while the fulfillment of the database with data about one system requires a great deal of time and effort, this effort can be used to design other systems as well. The database can be filled in such a way that different systems will partially share with each other tasks, abilities, requirements, abstract elements, links and sub-elements as well as their implementations, so the fulfillment of the database will take less time and effort for every next system.

While, in this work, many tables of the database that are responsible for the compatibility of elements of designed devices were filled manually, this process can be automated. For example, based on the content of different online shops that are selling controllers and components for the implementation of microcontroller-based devices, it is possible to fill the database with information about such implementations automatically with the help of the parsing script. In addition, with the help of a user-friendly interface, the task of fulfillment of the database can be shared among the community of enthusiasts.

The selection process can be improved with the use of genetic algorithms during the automated selection of implementations of different components and controllers among options that are satisfying given requirements. Based on priorities of parameters, like price, energy consumption and computation efficiency, it would be possible to solve the optimization task to find reasonable component compositions.

## 7. Conclusions

In this work, a new approach to the design of microcontroller-based physical security systems was presented, which allows combining various design techniques into the mostly automated solution with minimal involvement of the operator. This approach is modular and extensible, takes into account the security of the physical layer of designed systems, works with abstract system representation and looks for a trade-off between the security of the final solution and the resources expended on it. The methodology has a strong focus on security and aims at ensuring the protection of designed systems against various attacks at early stages of their lifecycle, while security components are considered as an integral part of the system.

For the experiment, we decided to design a system that provides perimeter monitoring based on mobile robots. The experiment was conducted with the help of the methodology software implementation. The developed methodology was evaluated in accordance with functional and non-functional requirements as well as compared with scientific and commercial solutions. Moreover, to obtain the average design time for each parameter combination of the attacker, the software implementation was executed 100 times for each combination of values on the computer with Windows 10 × 64 operating system, Intel Core i7-4790 CPU 3.60 GHz (8 cores) processor, 2 TB HDD and 32 GB RAM.

Attacker parameters define the attack actions that are possible for the attacker, which means that the amount of security elements integrated into the designed system differs from one combination of parameters to another. The time consumption was measured with the help of the time Python library. The minimum design time was 0.2941 s, while the maximum was 0.3408.

The results obtained in this work are very important for solving fundamental issues in the field of ensuring the information security of microcontroller-based systems. Moreover, the results of this work can be brought to practical use in the form of a software product. The use of such a product will help to reduce the amount of weak places and architectural defects in microcontroller-based systems, thereby, significantly reducing their attack surface. In turn, this will reduce the security risks that can lead to financial losses and losses of time as well as of the safety of people, which ensures the relevance and high significance of this work.

This indicates that, to prevent the possibility of implementation of attack actions in microcontroller-based systems, the developed methodology first analyses attack actions that are possible in accordance with parameters of the attacker and system component composition. After that, the methodology integrates security elements that are necessary to prevent those attack actions into the system devices, and thereby the system becomes protected from cyber-physical attacks.

It is important to note that the methodology is not aimed to replace security experts. In most situations, an expert in the security of microcontroller-based systems knows about the existing best and highly specialized solutions and is able to form alternatives at a very high level, while the quality of the solution provided by the methodology directly depends on the correctness and completeness of the database. However, it can be useful for an expert to automate routine tasks and provide alternative solutions.

In addition, the process of manual fulfillment of all tables of the developed database is required for the correct work of the software implementation. While the fulfillment of the database with data about one system requires a great deal of time and effort, this effort can be used to design other systems as well. The database can be filled in such a way that different systems will partially share tasks, abilities, requirements, abstract elements, links and sub-elements as well as their implementations with each other, and thus the fulfillment of the database will take less time and effort for every new system.

While, in this work, many tables of the database that are responsible for the compatibility of elements of designed devices were filled manually, this process can be automated. For example, based on the content of different online shops that are selling controllers and components for the implementation of microcontroller-based devices, it is possible to fill the database with information about such implementations automatically with the help of the parsing script.

This work presented not only the developed methodology for the design of microcontroller-based physical security systems with its software implementation but also a framework that can be improved in various ways.

For example, it can be improved with the use of genetic algorithms during the automated selection of implementations of different components and controllers among options that are satisfying given requirements. Based on priorities of parameters, like price, energy consumption and computation efficiency, it would be possible to solve the optimization task to find reasonable component compositions.

In addition, the verification process can become an integral part of the solution. It can provide the formal check of the possibility to design microcontroller-based physical security systems in accordance with the given requirements [24]. Moreover, it can provide the formal check of the security level of the designed system in accordance with the model of the attacker.

The use of a component-based approach to modeling microcontroller-based physical security systems can be extended with semi-natural, simulation and analytical modeling [48]. The advantage of integration of these approaches is in the possibility to represent various aspects of such systems, including dynamic ones.

## Figures and Tables

**Figure 1 sensors-21-08451-f001:**
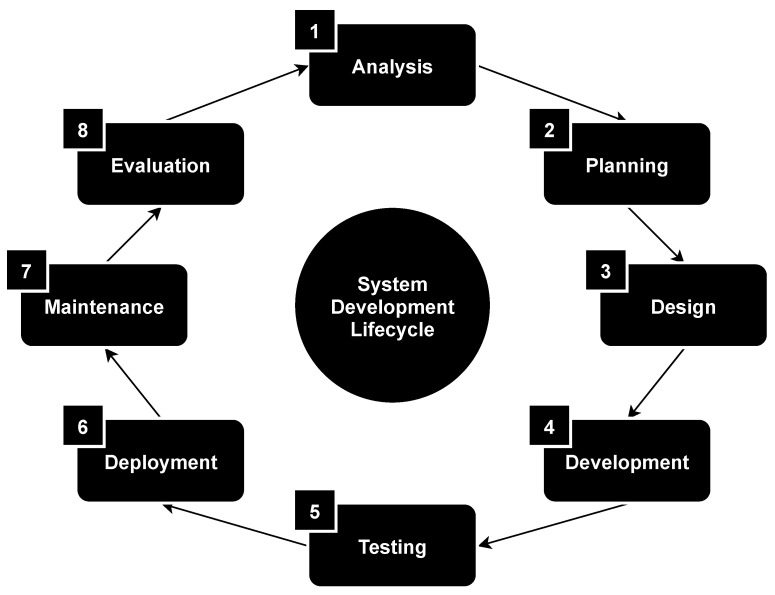
Microcontroller-based systems development lifecycle.

**Figure 2 sensors-21-08451-f002:**
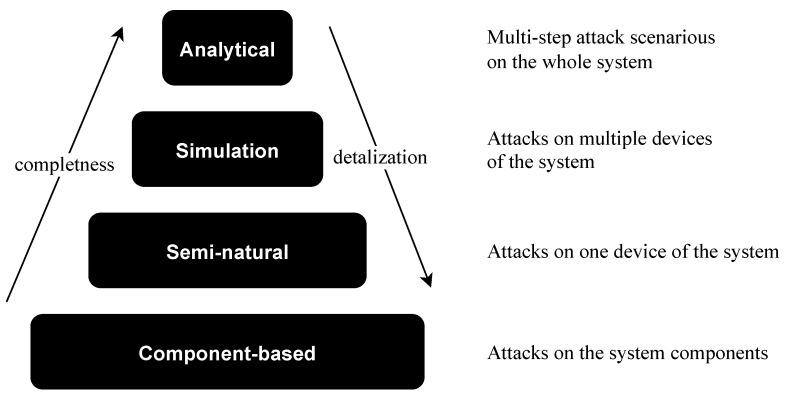
Comparison of modeling approaches.

**Figure 3 sensors-21-08451-f003:**
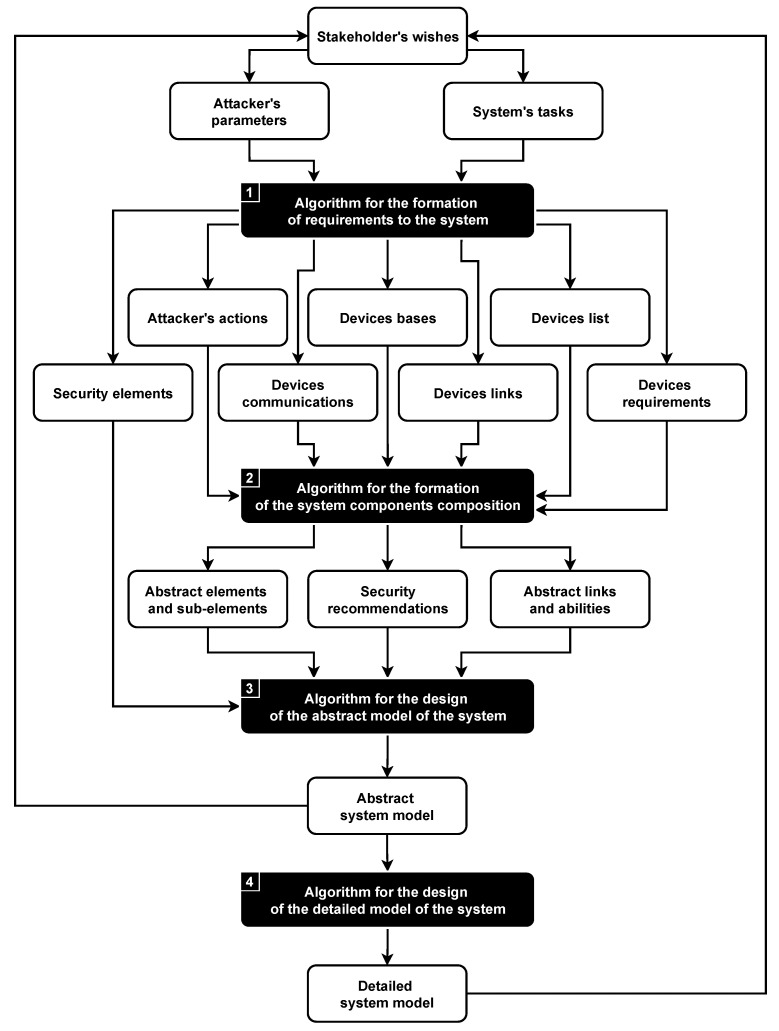
Methodology for the design of microcontroller-based physical security systems.

**Figure 4 sensors-21-08451-f004:**
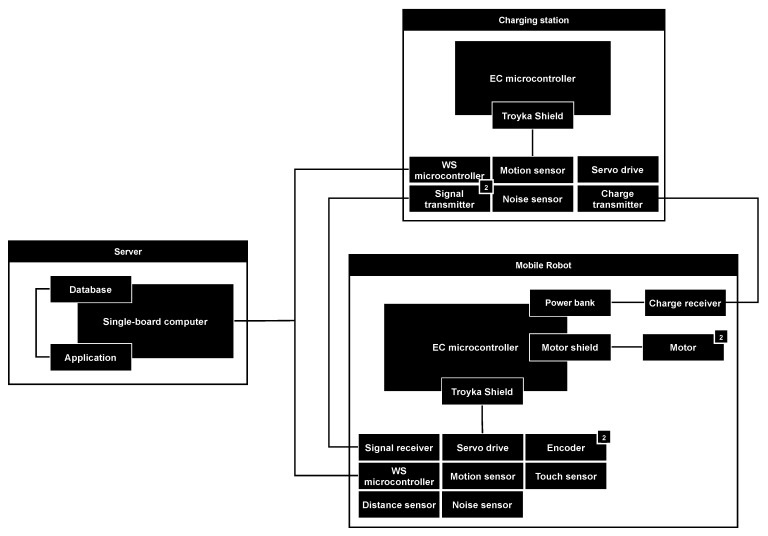
Architecture of the perimeter monitoring system.

**Figure 5 sensors-21-08451-f005:**
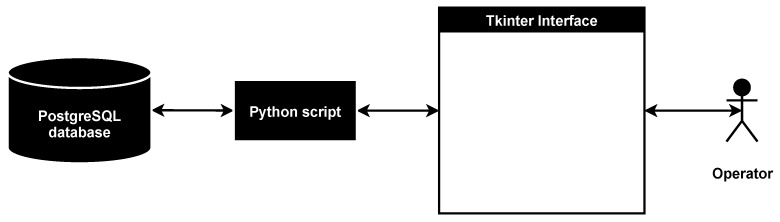
The architecture of the software implementation.

**Figure 6 sensors-21-08451-f006:**
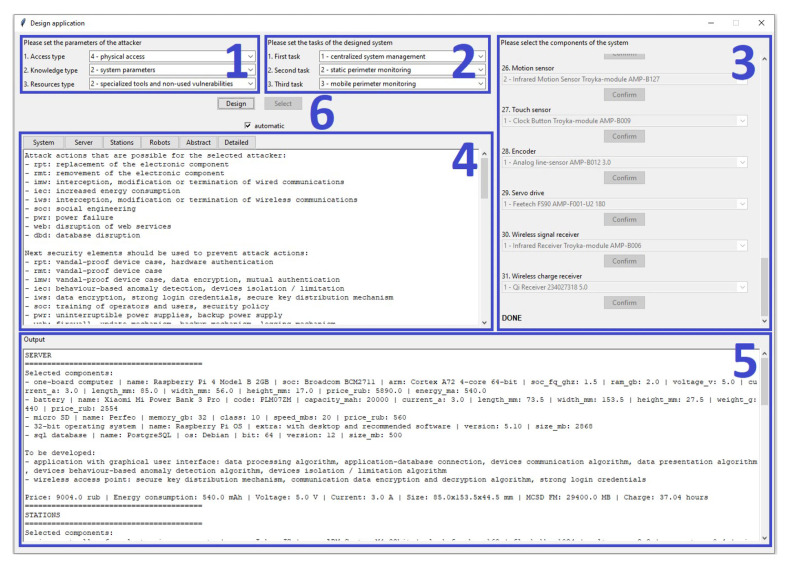
Interface of the application: state after the design process.

**Figure 7 sensors-21-08451-f007:**
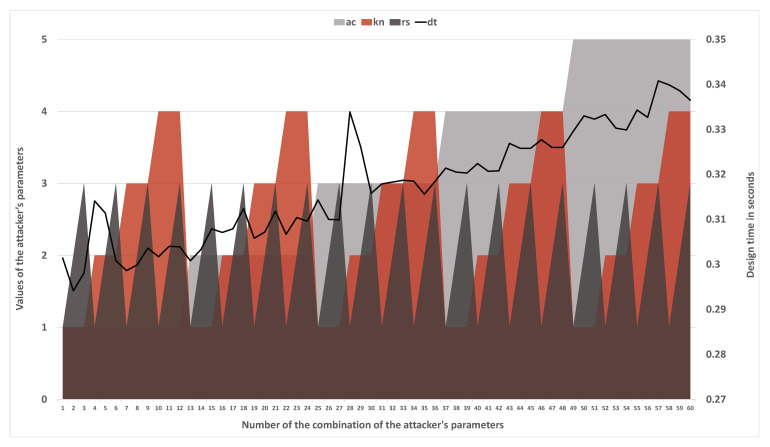
Dependencies between the design time and the attacker’s parameters.

**Table 1 sensors-21-08451-t001:** Various types of links between elements.

		*R*	*I*	*E*
$L_{mbs}$	Wi-Fi	IEEE 800.11	wireless 2.4 GHz	device ↔ device
	ZigBee	IEEE 802.15.4	wireless 2.4 GHz	
	Bluetooth	IEEE 802.15.1	wireless 2.4 GHz	
	nRF24L01+	ESB	wireless 2.4 GHz	
	Infrared	NEC	wireless 38 kHz	
$L_{bb}$	I2C	SDA + SCL	TWI	controller ↔ controller
	Serial	TxRx	UART	
	RS-232	RS232	UART	
	RS-485	RS485	UART	
$L_{hw}$	pin-to-pin	shared power	shield	controller ↔ component
	SVG	AVR I/O pin	three wires	
	VG	AVR I/O pin	two wires	
$L_{sw}$	method	functions	compiler	software ↔ software
	database	SQL queries	psycopg2	
	API	JSON structures	POST/GET	

**Table 2 sensors-21-08451-t002:** Attacker types of access.

	Description
1	No access to the system
2	Access to the system through global networks
3	Access to the system through local networks
4	Physical access to the system
5	Full access to the system

**Table 3 sensors-21-08451-t003:** Attacker types of knowledge.

	Description
1	General knowledge about the system from publicly available sources
2	Knowledge about parameters of the system
3	Knowledge about means of protection of the system
4	Knowledge about software and hardware of the system

**Table 4 sensors-21-08451-t004:** Attacker types of resources.

	Description
1	Widely-spread software tools and known vulnerabilities
2	Specialized software tools and previously non-used vulnerabilities
3	Possibility to investigate the system

**Table 5 sensors-21-08451-t005:** Classes of attack actions and different types of attackers.

			*a*											
			*ac*					*kn*				*rs*		
			1	2	3	4	5	1	2	3	4	1	2	3
*cl*	*cn*	gie			+	**+**	+		+	+	+			+
		bcd			+	+	+			+	+		+	+
		rpt				+	+		+	+	+	+	+	+
		rmt				+	+	+	+	+	+	+	+	+
	*cr*	rfw				+	+				+			+
		rbl				+	+				+			+
		mup			+	+	+			+	+		+	+
		imw				+	+	+	+	+	+	+	+	+
	*dv*	vau			+	+	+			+	+		+	+
		cad			+	+	+			+	+		+	+
		iec			+	+	+		+	+	+	+	+	+
		iws			+	+	+	+	+	+	+		+	+
	*st*	soc	+	+	+	+	+	+	+	+	+	+	+	+
		pwr				+	+	+	+	+	+	+	+	+
		web		+	+	+	+		+	+	+	+	+	+
		dbd		+	+	+	+		+	+	+	+	+	+

**Table 6 sensors-21-08451-t006:** Classes of attack actions and security elements.

			Security Elements
cl	cn	gie	anomaly detection algorithm, hidden placement of sensors
		bcd	events correlation algorithm, hidden placement of sensors
		rpt	vandal-proof device case, hardware authentication
		rmt	vandal-proof device case
	*cr*	rfw	vandal-proof device case, firmware encryption
		rbl	vandal-proof device case, bootloader encryption
		mup	vandal-proof device case, removal of physical update interface
		imw	vandal-proof device case, encryption, authentication
	dv	vau	strong login credentials, password policy, brute-force protection
		cad	strong encryption algorithms, secure key distribution mechanism
		iec	behavior-based anomaly detection, devices isolation/limitation
		iws	strong encryption algorithm on access point, strong login credentials, public key pair based authentication
	st	soc	training of operators and users, security policy
		pwr	uninterruptible power supplies, backup power supply
		web	firewall, update mechanism, backup mechanism, logging mechanism
		dbd	input validation, strict access policy, strong login credentials, separate database users for different operations

**Table 7 sensors-21-08451-t007:** Classes of attack actions and non-security elements.

			mbs
cl	cn	gie	sensors and receivers that react on the environment
		bcd	sensors that monitor environment
		rpt	any component
		rmt	any component
	cr	rfw	any controller with rewritable firmware
		rbl	any controller with rewritable bootloader
		mup	any controller with update system
		imw	controller ↔ controller, controller ↔ component
	dv	vau	device ↔ device, where authentication is used
		cad	device ↔ device, where encryption is used
		iec	devices with sleep mode/wireless interfaces
		iws	device ↔ device
	*st*	soc	any system with operators or/and users
		pwr	any system that relies on power grid
		web	any system with web-services
		dbd	any system with database

**Table 8 sensors-21-08451-t008:** Tasks, abilities and requirements of the designed system.

Task	Ability	Requirement	Dependency
centralized system management	to store and process system data	device that represents *the* *server* of the system	
	to run executable applications		
	to download and install software updates		
	to create wireless access points		
	to communicate with mobile robots		
	to communicate with charging stations		
	to provide user interface for operators of the system		
static perimeter monitoring	to provide wireless charging	devices that represent *charging stations* of the system	to provide *static* *perimeter monitoring*, the task of *centralized* *system management* should already be satisfied
	to monitor the perimeter nearby		
	to communicate with mobile robots		
	to communicate with the server of the system		
mobile perimeter monitoring	to be charged wirelessly	devices that represent *mobile robots* of the system	to provide *mobile* *perimeter monitoring*, the tasks of *centralized* *system management* and *static perimeter* *monitoring* should already be satisfied
	to navigate through the perimeter		
	to detect and chase intruders		
	to communicate with charging stations		
	to communicate with the server		
appropriate level of security	attackers with ac=4, kn=2, rs=2	security requirements should be taken into account during formation of all devices of the system	

**Table 9 sensors-21-08451-t009:** Tasks, abilities and requirements related to the server.

Task	Ability	Requirement	Dependency
work cycle support	to store data	32-bit operating system	
		sql database	
	to update software	wire network interface	
		software update server	
		software update mechanism	
	to run applications	32-bit operating system	
	to create wireless access points	32-bit operating system	
		wireless network interface	
		access points configuration mechanism	
interaction with operators	to provide graphical user interface	application with GUI	to provide *interaction* *with operators*, the task of *work cycle support* should already be satisfied
		app-db connection	
		data processing algorithm	
		data presentation algorithm	
interaction with other devices	to communicate with other devices	wireless network interface	to provide *interaction* *with other devices*, the task of *work cycle* *support* should already be satisfied
		devices communication algorithm	
appropriate level of security	attackers with ac=4, kn=2, rs=2	security should be taken into account during the design of all elements of the device	

**Table 10 sensors-21-08451-t010:** Tasks, abilities and requirements related to the charging stations.

Task	Ability	Requirement	Dependency
work cycle support	to update firmware	wireless network interface	
		bootloader	
		firmware update mechanism	
	to charge parked devices	wireless charge transmitter	
interaction with intruders	to detect intruders	motion sensor	to provide *interaction* *with intruders*, the task of *work cycle support* should already be satisfied
		noise sensor	
		servo drive	
		intruder detection algorithm	
interaction with parking devices	to help mobile devices to park near	wireless signal transmitter	to provide *interaction* *with parking devices*, the task of *work cycle* *support* should already be satisfied
		parking direction algorithm	
interaction with the server	to communicate with the server	wireless network interface	to provide *interaction* *with the server*, the task of *work cycle* *support* should already be satisfied
		server communication algorithm	
appropriate level of security	attackers with ac=4, kn=2, rs=2	security should be taken into account during the design of all elements of the device	

**Table 11 sensors-21-08451-t011:** Tasks, abilities and requirements related to the mobile robots.

Task	Ability	Requirement	Dependency
work cycle support	to update firmware	wireless network interface	
		bootloader	
		firmware update mechanism	
	to be charged in a wireless way	wireless charge receiver	*battery* should provide power supply for *8 h*
		battery	
		charge monitoring algorithm	
perimeter monitoring	to move	collector motor	*to move*, the *work cycle support* task should already be satisfied
		movement algorithm	
	to avoid obstacles	distance sensor	*to avoid obstacles*, each robot should already have an ability *to move*
		touch sensor	
		servo drive	
		obstacles detection algorithm	
		obstacles avoidance algorithm	
	to navigate	encoder	*to navigate*, each robot should already have and ability *to avoid* *obstacles*
		map construction algorithm	
		path construction algorithm	
interaction with intruders	to detect intruders	motion sensor	*to detect intruders*, the *perimeter monitoring* task should already be satisfied
		noise sensor	
		servo drive	
		intruders detection algorithm	
	to chase intruders	distance sensor	*to chase intruders*, each mobile robot should already have an ability *to detect intruders*
		intruders chase algorithm	
interaction with charging stations	to park near charging stations	wireless signal receiver	*to park near charging* *stations*, the *perimeter* *monitoring* task should already be satisfied
		parking algorithm	
interaction with the server	to communicate with the server	wireless network interface	*to communicate with* *the server*, the *work* *cycle support* task should already be satisfied
		server communication algorithm	
appropriate level of security	attackers with ac=4, kn=2, rs=2	security should be taken into account during the design of all elements of the device	

**Table 12 sensors-21-08451-t012:** The comparison with commercial and scientific solutions.

	Solutions	Levels of the System				Classes of Attack Actions			
		cn↔cr	cr↔cr	dv↔dv	st↔st	cn	cr	dv	st
**Scientific**	[13]	–	–	–	–	–	*	*	–
	[14]	–	–	–	–	–	*	*	–
	[9]	+	+	–	–	+	+	–	–
	[10]	+	+	–	–	–	+	–	–
	[12]	+	+	–	–	+	+	–	–
	[15]	–	–	+	–	–	+	+	–
**Commercial**	[16]	–	–	–	+	–	–	+	+
	[17]	–	–	–	+	–	–	+	+
	[18]	–	–	–	+	–	–	+	+
	[19]	–	–	–	+	–	–	–	+
	[20]	–	–	–	+	–	–	+	+
	[21]	–	–	–	+	–	–	–	+
**Developed**		+	+	+	+	+	+	+	+

Note that “*” for [13,14] means that provided models and approaches can be improved for taking the corresponding classes of attack actions into account.

## Data Availability

The software implementation of the presented design methodology is available for download: https://github.com/levshun/PhD-mcbpss_design (accessed on 5 December 2021).

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
