# Peer review of "Design of Secure Microcontroller-Based Systems: Application to Mobile Robots for Perimeter Monitoring"

_sensors, 2021, doi:10.3390/s21248451_

Round 1

Reviewer 1 Report

The content of the article is consistent with the scientific area of the journal Sensors. 
The subject raised by the authors is current in this area. 
This article describes a design of secure microcontroller-based systems.
The paper has an original and research nature.
For a better clarification, please edit your paper as follows: 
* Enlarge the Introduction about general information about topic with current results reported in the world and Europe - References to expand the results of European authors registered in WoS, such as 
Mobile Robot Controlling Possibilities of Inertial Navigation System
Analysis of control and correction options of mobile robot trajectory by an inertial navigation system 

* Conclusions and future work should be extended to contain practical applications based on research described in this paper, 

* Figure 5 - axis description is missing
* It is necessary to add the used equations to the scientific article
* The title of chapters 3.4 and 3.5 does not correspond to the text in Figure 2

I recommend publishing the post after the proposed modifications.

Author Response

Dear reviewer, 

For more details, please, check the new version of the paper. All additions/reconsiderations are highlighted in yellow.

Reviewer 2 Report

The authors have presented an automated methodology and software tool for analysis of security properties of micro-controller-based networked systems. The tool provides information about security requirements, possible attacks, etc. An example scenario with mobile robots is also provided.

The manuscript is well-written and easy to follow. Details of the methodology have been explained clearly.

This is great engineering work. However, research aspect of this work is lacking. The paper is almost in the form of a datasheet / user manual for the proposed software tool. Also, major portion of the research background is derived from the authors' previous work [23, 24].

It will be useful to discuss technical aspects of the security requirements of such networks, how the tool can be used to prevent attacks, etc.

Author Response

(The authors gave the same response as above.)

Round 2

Reviewer 2 Report

The paper has been well revised to address previous review comments. Addition of the new section along with other clarifications have improved the manuscript.